# A Non-Convex Compressed Sensing Model Improving the Energy Efficiency of WSNs for Abnormal Events’ Monitoring

**DOI:** 10.3390/s22218378

**Published:** 2022-11-01

**Authors:** Yilin Huang, Haiyang Li, Jigen Peng

**Affiliations:** School of Mathematics and Information Science, Guangzhou University, Guangzhou 510006, China

**Keywords:** wireless sensor network, monitoring abnormal events, energy-efficient compressed data gathering model, compressed sensing, alternating direction iteration fraction penalty thresholding algorithm

## Abstract

The wireless sensor network (WSN), a communication system widely used in the Internet of Things, usually collects physical data in a natural environment and monitors abnormal events. Because of the redundancy of natural data, a compressed-sensing-based model offers energy-efficient data processing to overcome the energy shortages and uneven consumption problems of a WSN. However, the convex relaxation method, which is widely used for a compressed-sensing-based WSN, is not sufficient for reducing data processing energy consumption. In addition, when abnormal events occur, the redundancy of the original data is destroyed, which makes the traditional compressed sensing methods ineffective. In this paper, we use a non-convex fraction function as the surrogate function of the $\ell_{0}$-norm, which achieves lower energy consumption of the sensor nodes. Moreover, considering abnormal event monitoring in a WSN, we propose a new data construction model and apply an alternate direction iterative thresholding algorithm, which avoids extra measurements, unlike previous algorithms. The results showed that our models and algorithms reduced the WSN’s energy consumption during abnormal events.

## 1. Introduction

Wireless sensor network (WSN), an important communication system in the Internet of Things, is widely used in fields such as the environment, medical care, transportation, and security [1,2]. Its basic structure consists of a large number of wireless sensor nodes and a sink: the nodes collect and transmit data to the sink, and the sink summarizes and processes the data. The energy of wireless sensor nodes usually depends on batteries or solar power, while the energy reserve of the sink is usually sufficient, which makes the energy consumption of wireless sensor nodes one of the key issues in a WSN [3]. In recent years, with the emergence of compressed sensing theory [4], signal processing methods in a WSN based on sparse representation have already attracted widespread attention [5,6,7,8]. Generally speaking, signal acquisition in a WSN can be expressed as a linear measurement:(1)y=Φd,
where *y* is the measurement value obtained by the wireless sensor node sampling; *d* is the original data; the measurement matrix Φ with a size of M×N is an encoding process of the original data *d*. It is assumed in compressed sensing theory that the data are redundant, which can be sparsely represented by a transform Ψ satisfying d=Ψx, where *x* is a sparse vector. The compressed sensing theory shows that *d* can be reconstructed by solving an optimization model. It is worth noting that sampling is operated by the nodes and that reconstruction is operated by the sink, which means that compressed sensing transfers some workload from the nodes to the sink, and it is suitable for the energy distribution characteristic of a WSN. Additionally, when the WSN is monitoring abnormal events, the sparsity of signals will be disturbed, i.e., signals are no longer sparse in the original transform domain, which makes it impossible to reconstruct signals using the original compressed sensing model. In general, the so-called abnormal events appear sparsely and sporadically; hence, they are naturally sparse in a WSN. In [6], a feasible approach was proposed that regards the original signal as the summation of the normal and abnormal events’ signals: the normal signal is sparse under the sparse transform domain, and the abnormal events’ signal is sparse under the identical transform domain, so the splicing signal in this kind of transform (Ψ|I) is sparse.

The studies [9,10,11,12] discussed the feasibility and application of compressed sensing in signal acquisition and processing, demonstrating that a WSN can effectively reduce network energy consumption by using compressed sensing. The compressed sensing theory was applied to a WSN for the first time, and in [5], the concept of compressed wireless sensing (CWS) was proposed and found to be effective for energy efficiency in a WSN of a certain structure. In [6], a compressive data gathering (CDG) model was proposed, which opened a new path for data gathering. In [7], the authors discussed the feasibility of using a binary matrix as the measurement matrix to improve CDG, where the wireless sensor nodes did not need multiplication, further reducing computational consumption. Based on CDG, the researchers [8] developed a distributed compressed sensing method by dividing the data into several blocks for reconstruction with the idea of dividing-and-conquering and then summarizing the data to the sink.

Although compressed sensing theory and its improvements have achieved excellent results, there are still problems with the models in the above-mentioned research:

*(1)* These studies of compressed sensing models usually use the $\ell_{1}$-norm as the convex surrogate form of the $\ell_{0}$-norm. Even though it is typical and popular [13,14], the $\ell_{1}$-norm often fails to produce the sparsest solution or select the position of the non-zero coefficient [15,16], which implies that it may not be the best choice for a WSN. In fact, many studies have shown that non-convex surrogate functions achieve a better approximation than the $\ell_{0}$-norm and a lower sampling rate than the $\ell_{1}$-norm [15,16,17,18].

*(2)* A conventional and effective approach to focusing on abnormal events’ monitoring in a WSN is to treat abnormal events as sparse vectors under identity transformation, just as [6] did, to reconstruct additional abnormal event signals with the same length as normal signals. However, this method requires the reconstruction of a signal with twice the dimension of the original signal, which means more measurements and energy consumption.

This paper mainly discusses the two problems mentioned above. First, we adopted the non-convex fraction function $P_{a}\left(x\right)$ as the surrogate form of the $\ell_{1}$-norm in the compressed sensing model, where
(2)$$P_{a}\left(x\right)=\sum_{i=1}^{N}p_{a}\left(x_{i}\right)=\sum_{i=1}^{N}\frac{a\left|x_{i}\right|}{1+a\left|x_{i}\right|}\;\left(a\geq 0\right).$$

Following estimates of the measurement number *M* in compressed sensing theory, we thought that the non-convex fraction function model could decrease the number of measurements significantly compared with the $\ell_{1}$-norm model, implying that the wireless sensor nodes in our model have a lighter workload and consume less energy. Second, regarding abnormal events’ monitoring, we propose an optimization model based on the non-convex fraction function and an alternate direction iterative fraction penalty thresholding (AD-IFPT) algorithm to solve it. In each iteration, the normal and abnormal events signals were all *N*-dimensional and alternately reconstructed. Meanwhile, we analyzed the AD-IFPT algorithm and verified its convergence. Based on the minimal number of measurements theory in compressed sensing and the results of our numerical experiment, we concluded that the AD-IFPT algorithm requires fewer measurements compared with other traditional approaches, meaning that its energy efficiency in abnormal events’ monitoring is better than that of other methods.

The rest of this paper is organized as follows. In Section 2, we briefly introduce the theory of compressed sensing and its applications in a WSN. In Section 3, we introduce a compressed sensing optimization model based on a specific non-convex surrogate function, the fraction function. Then, focusing on abnormal events’ monitoring, we designed an alternating direction algorithm, which has lower energy consumption and high efficiency in abnormal events’ monitoring. We also give several conclusions about the properties of the proposed algorithm. The results of the simulation experiments are presented in Section 4. Finally, we give our conclusion in Section 5.

## 2. Preliminary Knowledge

As described above, compressed sensing theory can reduce the energy consumption of signal acquisition and transmission, but increase the energy consumption of signal decompression, which just meets the requirements of low-energy consumption at wireless sensor nodes and unlimited energy consumption at the sink in a WSN. In this section, we introduce the compressed sensing theory and the definition of abnormal events to be able to discuss the proposed model clearly in the next section and to choose two applications of compressed sensing theory to evaluate the performance of the proposed model in the experiment section.

### 2.1. Compressed Sensing Theory

In traditional signal processing, obtaining an *N*-dimensional signal requires at least *N* measurements. However, compressed sensing theory shows that, if an original signal *d* can be sparsely represented, that is, if there exists a sparse transform Ψ such that
(3)d=Ψx,
where *x* is a sparse vector, meaning that most entries of *x* are zero, then we can reconstruct the original signal *d* with fewer measurement times *M* by solving the following optimization problems:(4)$$\left(P_{0}\right)\;\min_{x}{∥x∥}_{0}\;\;s.\;t.\;\;\;y=\Phi \Psi x,$$
where *y* is the measurement value and Φ is the measurement matrix with the size of M×N and M≪N. Because of the discrete and discontinuous properties of the $\ell_{0}$-norm, the $P_{0}$ problem is actually nondeterministic polynomial time (NP) hard [19]. Moreover, the $\ell_{0}$-norm is very sensitive to noise, which makes the model $\left(P_{0}\right)$ problematic for solving this problem. To overcome these difficulties, we replaced the highly discontinuous $\ell_{0}$-norm with a continuous surrogate function [13,14,18,20,21,22,23], in which the $\ell_{1}$-norm is typical and popularly proposed. The constrained problem corresponding to the $\ell_{1}$-norm is
(5)$$\left(P_{1}\right)\;\min_{x}{∥x∥}_{1}\;\;s.\;t.\;\;\;y=\Phi \Psi x.$$

Given a signal with noise, the corresponding regularization problem is
(6)$$\left(P_{1}^{\lambda }\right)\;\min_{x}{\;\lambda ∥x∥}_{1}+{∥\Phi \Psi x-y∥}_{2}^{2},$$
where λ>0, the regularized parameter, represents a trade-off between reconstruction error and sparsity. It is worth noting that ($P_{1}$) and ($P_{1}^{\lambda }$) are convex optimization problems, which can be solved by many methods such as basis pursuit [24], $\ell_{1}$-magic [25], an iterative soft thresholding algorithm [4,26], and split Bregman methods [27,28].

In fact, some studies have shown that non-convex surrogate functions can obtain a lower sampling rate than the $\ell_{1}$-norm [29,30], which is extremely important in a WSN because it means that there is a lower energy consumption of signal acquisition and transmission at the nodes. In Section 3, we discuss the energy consumption of signal acquisition and transmission at the nodes and the abnormal events’ monitoring of a WSN by taking a specific non-convex surrogate function, the fraction function.

### 2.2. Analog Compressed Sensing

Using compressed sensing to measure analog signals has been applied in many fields [9,10,11,12]. The main idea is to obtain *M* measured values of an *N*-dimensional signal in an analog domain based on the random demodulator (RD) architecture (see Figure 1). The RD consists of three stages: mixer, integrator, and the analog-to-digital converter (ADC). In this case, there are three steps to signal acquisition:The mixer multiplies an input signal by a random sequence.The integrator accumulates the output voltage of the mixer.The ADC samples it at a rate of 1/*M*.

Step 1 corresponds to the linear measurement represented by the inner product of the original signal *x* with each row of the measurement matrix Φ. Step 2 gives the measurement values, and the sampling rate of Step 3 is lower than the Nyquist rate when M≪N, which implies that it will use less energy compared to traditional signal sampling methods. To implement compressed sensing, a prerequisite is that analog signals be sparsely represented as (Equation 3). In applications, for example, natural images can usually be sparsely represented under the discrete cosine transform (DCT) (see Figure 2), and [2] showed that the ocean temperature measured by a WSN can be sparsely represented under a discrete wavelet transform (DWT).

### 2.3. Compressed Data Gathering in a Multi-Hop WSN

In [6], a CDG model was proposed, which applies compressed sensing to a multi-hop WSN. In the traditional wireless sensor signal transmission model, each sensor node collects signals and transmits them to the nearest node in the direction of the sink. If the sensor receives signals from other sensors, those will also be sent to the next node. With the increase in sensor network scale, the power consumption of this data transmission mode will increase greatly. Moreover, the energy consumption between sensors is extremely uneven because, the closer a sensor is to the sink, the more signals it receives and the more transmission energy it consumes.

To overcome the poor performance of the traditional multi-hop WSN architecture, the CDG model was proposed based on compressed sensing theory [6]. In this model, each node only needs to multiply its reading by a random number, add it to the upstream node reading, and transmit the result to the next node, which avoids extra transmitting of data in the multi-hop architecture. During data sampling, the entire wireless sensor network needs to repeat the above steps *M* times and the data received at the sink can be represented by
$$\left(\begin{array}{c} y_{1}\\ y_{2}\\ \vdots \\ y_{M} \end{array}\right)=\left(\begin{array}{cccc} \Phi_{11} & \Phi_{12} & \ldots & \Phi_{1N}\\ \Phi_{21} & \Phi_{22} & \ldots & \Phi_{2N}\\ \vdots & & \vdots & \\ \Phi_{M1} & \Phi_{M2} & \ldots & \Phi_{MN} \end{array}\right)\left(\begin{array}{c} d_{1}\\ d_{2}\\ \vdots \\ d_{N} \end{array}\right),$$
where $y_{i}$ represents the *i*th measurement readings of the sink with M≪N; $\Phi_{ij}$ represents the random coefficient of the *j* node in the *i*th measurement; $d_{j}$ represents the readings of the *j* node for i=1,2,…,M. We set d=Ψx as the sparse representation of *d*, which made this model the same as the compressed sensing model (Equation 4). Its working model is shown in Figure 3.

### 2.4. Abnormal Events’ Monitoring

WSNs are widely used for monitoring abnormal events. For example, when a WSN collects temperature information in a forest, abnormally high readings may indicate a fire. In this case, when abnormal readings appear on nodes, the sparsity of the data will disappear, and the data can no longer be sparsely represented under the original sparse transform (see Figure 4). If it is determined that abnormal events do not appear on a large scale, then the abnormal readings are considered to be sparse in all node readings in the WSN. For this reason, assuming the original signal contains both normal and abnormal events, we express the sensor readings as
(7)$$d=d_{0}+d_{s},$$
where $d_{0}$ represents the original readings and $d_{S}$ represents the abnormal readings. Since abnormal readings are sparse and the original signal is sparse under the sparse transform Ψ, then
(8)$$y=\Phi d,\;d=d_{0}+d_{s}=\Psi x_{0}+Ix_{s},$$
where $d_{0}=\Psi x_{0}$; $d_{s}=x_{s}$; $x_{0}$ and $x_{s}$ are sparse vectors. A reasonable choice was given in [6], which combines the matrices Φ, Ψ, and *I* as a new measurement matrix of the following form:(9)$$\begin{array}{cc} {\min∥x∥}_{0}\;\;s.\;t.\;\; & y=\Phi d,d=\left(\Psi \;I\right)x,\\ \; & x={\left(x_{0}^{T}\;x_{s}^{T}\right)}^{T}. \end{array}$$

If $\tilde{\Psi }=\left(\Psi \;I\right)$, then it is the original compressed sensing model as in (Equation 4). However, the dimension of the signal is doubled, which presents a challenge for reconstructing a signal with abnormal readings using fewer measurements. In the following discussion, we propose an alternative reconstruction method and an algorithm for abnormal event monitoring that does not need to solve the original signal with twice the length in the reconstruction process.

## 3. Non-Convex Compressed Sensing Model in a WSN and Alternating Direction Reconstruction Algorithm

In this section, we first introduce a compressed sensing optimization model based on a specific non-convex surrogate function, the fraction function. Second, focusing on abnormal events’ monitoring, we propose an alternating direction algorithm that has lower energy consumption and high efficiency.

### 3.1. Fraction Function Abnormal Events Monitoring Model in WSN

In this part, we mainly introduce our proposed model, but first, we discuss the estimation of the minimum number of measurements in different compressed sensing models: the number *M* of rows of the measurement matrix. The convex relaxation model (Equation 6) is the most widely used and derives its optimal solution easily because of the excellent properties of the convex function. According to the existing conclusions of compressed sensing theory, for the estimation of the minimum number of measurements in (Equation 6), the following inequalities hold:(10)M≥CKln(N/K),
where *N* is the number of columns of the measurement matrix; *K* is the sparsity level of the original data; *C* is a constant number [31]. In applications, *M* is directly related to the sampling times of the sensor networks. To achieve better energy consumption, a lower bound on *M* is required. Since the energy at the sink is sufficient, the additional effort for the reconfiguration is negligible. Therefore, we sacrificed the excellent performance of the convex function in optimization to choose the fraction non-convex function as a surrogate function to achieve better performance for the minimum number of measurements.

Let the fraction penalty compressed sensing model be set $\left(P_{P_{a}}\right)$ as
(11)$$\left(P_{P_{a}}\right)\;\min_{x}P_{a}\left(x\right)\;\;s.\;t.\;\;\;y=\Phi d,d=\Psi x,$$
where $P_{a}\left(x\right)$ is defined in (Equation 2). Although there is no rigorous theoretical proof, numerical experiments showed that the minimum number of measurements required for a non-convex fraction function is lower than for convex functions in compressed sensing, which implies that it can recover data with less sampling. The comparison of the fraction penalty function with another non-convex function the $\ell_{0.5}$-norm was explained in [17] in detail, which showed that the fraction penalty function has better sparse approximation and a lower number of measurements than the $\ell_{0.5}$-norm. The better properties of the fraction penalty function are helpful to achieve better energy efficiency in a WSN.

After having introduced abnormal events’ monitoring in a WSN, let us now give its fraction penalty form for abnormal events’ reconstruction. We assumed that the WSN signal *d* could be split into a normal signal $d_{0}$ and an abnormal events–readings vector $d_{S}$ as in (Equation 7). $d_{0}$ has its sparse representation $d_{0}=\Psi x_{0}$, and $d_{S}$ sparsely appears in the WSN such that $d_{S}=Ix_{s}$ as (Equation 8), where $x_{0}$ and $x_{s}$ are $K_{1}$-sparse and $K_{2}$-sparse vectors, respectively. Thus, the abnormal events’ monitoring based on fraction compressed sensing is
(12)$$\begin{array}{cc} \min_{x_{0},x_{s}}P_{a}\left(x_{0}\right)+P_{a}\left(x_{s}\right)\;\;s.\;t.\;\; & y=\Phi d,\\ \; & d=\Psi x_{0}+x_{s}, \end{array}$$

Next, we focus on the algorithm design of Problem (Equation 12).

### 3.2. Algorithm Design

Before we discuss the solution of the problem (Equation 14), we first discuss the shortcomings of the solution of such problems in previous studies. In [6], an available method was to splice the sparse transform and identity matrices into an M×2N matrix, which regards the abnormal readings as a sparse vector under the identity transform. Hence, within abnormal events’ monitoring, it took the form:   
(13)$$\begin{array}{c} \min_{x}\left\{∥\Phi \left(\Psi |I\right){x-y∥}_{2}^{2}+\lambda P_{a}\left(x\right)\right\}, \end{array}$$
where $x={\left(x_{0}^{T}\;x_{S}^{T}\right)}^{T}$. It is worth noting that *x* in this model is a 2N-dimensional signal, which implied that we needed to reconstruct a 2N-dimensional signal with *M* measurements. In addition, the sparsity of the reconstructed vector *x* became $K=K_{1}+K_{2}$. Based on the suggestion of a numerical simulation experiment, the increase in the signal dimension and sparsity led to a higher measurement number and more WSN energy consumption.

Now, let us introduce our approach. To solve (Equation 12), we considered its corresponding regularization problem:(14)$$\begin{array}{c} \min_{x}\left\{∥\Phi \Psi x_{0}+\Phi x_{s}{-y∥}_{2}^{2}+\lambda_{1}P_{a}\left(x_{0}\right)+\lambda_{2}P_{a}\left(x_{s}\right)\right\}, \end{array}$$
where $\lambda_{1}$ and $\lambda_{2}$ are regularization parameters, which are used to trade off the sparsity weight. Inspired by the idea of the alternating direction iteration algorithm, we propose an approach to reconstruct the original signal and abnormal events’ vectors alternately. In each iteration, the iterative fraction penalty thresholding algorithm is used to solve the optimization problem (Equation 15) or (Equation 16). More specifically, we considered the following two optimization problems:(15)$$\begin{array}{c} \min_{x_{0}}\left\{∥\Phi \Psi x_{0}+\Phi x_{s}{-y∥}_{2}^{2}+\lambda_{1}P_{a}\left(x_{0}\right)\right\}, \end{array}$$
and
(16)$$\begin{array}{c} \min_{x_{s}}\left\{∥\Phi \Psi x_{0}+\Phi x_{s}{-y∥}_{2}^{2}+\lambda_{2}P_{a}\left(x_{s}\right)\right\}. \end{array}$$

To solve (Equation 15), we set $x_{s}$ to be fixed and did the same to solve (Equation 16). Notice that (Equation 15) and (Equation 16) have a similar form, which could be made equivalent by a simple substitution. To solve (Equation 15), we set $\hat{y}=y-\Phi x_{S}$; similarly, to solve (Equation 16), we set $y_{0}=y-\Phi \Psi x_{0}$. Therefore, we only needed to consider the following suboptimization problem:(17)$$\left(P_{P_{a}}^{\lambda }\right)\;\min_{x}\;\lambda P_{a}\left(x\right)+{∥\Phi \Psi x-y∥}_{2}^{2}$$
and then alternate iterations through the above substitution form.

We considered using the thresholding algorithm to solve the regularization problem, because in [17], it was proven that the optimal solution $x^{*}=\left\{x_{1}^{*},x_{2}^{*},\ldots ,x_{N}^{*}\right\}$ to $P_{P_{a}}^{\lambda }$ could be represented by the following threshold operator form:(18)$$x_{i}^{*}=H_{\lambda \mu ,t}\left({\left(z_{\mu }\left(x^{*}\right)\right)}_{i}\right),$$
where *t* is a positive thresholding parameter; μ satisfies ${0<\mu <∥\Phi \Psi ∥}_{2}^{-2}$; $z_{\mu }\left(x^{*}\right)=x^{*}+\mu {\left(\Phi \Psi \right)}^{T}\left(y-\Phi \Psi x^{*}\right)$; the thresholding operator *H* has the following form:(19)$$H_{\lambda ,t}\left(x\right)=\left(H_{\lambda ,t}\left(x_{1}\right),H_{\lambda ,t}\left(x_{2}\right),\ldots ,H_{\lambda ,t}\left(x_{N}\right)\right),$$
(20)$$H_{\lambda ,t}\left(x_{i}\right)=\left\{\begin{array}{cc} h_{\lambda }\left(x_{i}\right), & \left|x_{i}\right|>t\\ 0, & \;\mathrm{otherwise}\; \end{array}\right.$$
(21)$$h_{\lambda }\left(x_{i}\right)=\mathrm{sgn}\left(x_{i}\right)\left(\frac{\frac{1+ax_{i}}{3}\left(1+2\cos\left(\frac{G(x_{i})}{3}-\frac{\pi }{3}\right)\right)-1}{a}\right),$$
and
(22)$$G\left(x_{i}\right)=\arccos\left(\frac{27\lambda a^{2}}{4(1+a|x_{i}{\left|\right)}^{3}}-1\right).$$

Hence, in the algorithm design, we had the following iteration form of *x*:(23)$$x^{k+1}=H_{\lambda \mu }\left(\left(z_{\mu }\left(x^{k}\right)\right)\right).$$

To sum up, we set the iterative scheme in the thresholding algorithm as follows:(24)$$\begin{array}{cc} \mathrm{Update} & \;x_{0}^{k+1}:\\ \mathrm{set} & \;y_{s}^{k}=y-\Phi x_{s}^{k}\\ & {\left(x_{0}^{k+1}\right)}_{i}=H_{\lambda_{1}u_{1},t}\left({\left(z_{u_{1}}\left(x_{0}^{k}\right)\right)}_{i}\right), \end{array}$$
(25)$$\begin{array}{cc} \mathrm{Update} & \;x_{s}^{k+1}:\\ \mathrm{set} & \;y_{0}^{k}=y-\Phi \Psi x_{0}^{k+1}\\ & {\left(x_{s}^{k+1}\right)}_{i}=H_{\lambda_{2}u_{2},t}\left({\left(z_{u_{2}}\left(x_{s}^{k}\right)\right)}_{i}\right). \end{array}$$

Based on the above iterative scheme, the alternating direction fraction penalty algorithm (AD-IFPT) is given as Algorithm 1. In this alternating direction iteration model, each step solves a compressed sensing optimization problem of reconstructing an *N*-dimensional vector from *M* measurements. This makes up the deficiency of (Equation 9), which doubles the signal dimension because of matrix splicing. In the subsequent simulation experiment, we saw that the AD-IFPT was effective and decreased the measurements more compared to the IFPT or traditional compressed sensing methods.
**Algorithm 1** Alternating direction iteration fraction penalty thresholding algorithm.**Require:***A*, *y*, $\lambda_{1}$, $\lambda_{2}$, $x_{0}^{0}$, $x_{s}^{0}$, $u_{1}$, $u_{2}$, *t*, and *a*
 If no convergence, do:
 $y_{0}^{n}=y-\Phi x_{s}$

 $z^{n}=x_{0}^{n}+u_{1}{\left(\Phi \Psi \right)}^{T}\left(y_{0}^{n}-\Phi \Psi x_{0}^{n}\right)$
 **for**
i=1 :length $\left(x_{0}\right)$ **do**

  **if** $\left|z_{i}^{n}\right|>t$ **then**
   $x_{0}^{n+1}=g_{\lambda_{1}u_{1}}\left(z_{i}^{n}\right)$
  **else**
   $x_{0}^{n+1}=0$
  **end if**
 **end for**
 $y_{S}^{n}=y-\Phi \Psi x_{0}$
 ${\tilde{z}}^{n}=x_{s}^{n}+u_{2}\Phi^{T}\left(y_{s}^{n}-\Phi x_{s}^{n}\right)$
 **for**
i=1 :length $\left(x_{s}\right)$ **do**
  **if** $\left|{\tilde{z}}_{i}^{n}\right|>t$ **then**
   $x_{s}^{n+1}=g_{\lambda_{2}u_{2}}\left({\tilde{z}}_{i}^{n}\right)$
  **else**
   $x_{s}^{n+1}=0$
  **end if**
 **end for**

 **return**
$x_{0}^{n+1},x_{s}^{n+1}$


### 3.3. Parameter Setting and Convergence Analysis

Next, we analyzed the convergence of the AD-IFPT. Beforehand, we needed to make use of a few auxiliary functions and introduce several lemmas that will be helpful in proving convergence.

**Lemma** **1**([17]). *Given $v\in R^{N}$, the following equation holds:*
(26)$$H_{\lambda ,t}\left(v\right)=\arg\min_{x}{∥x-v∥}_{2}^{2}+\lambda P_{a}\left(x\right),$$
*where t satisfies*
(27)$$t=\left\{\begin{array}{cc} \frac{\lambda a}{2}, & \;\mathrm{if}\;\lambda \leq \frac{1}{a^{2}}\\ \sqrt{\lambda }-\frac{1}{2a}, & \;\mathrm{if}\;\lambda >\frac{1}{a^{2}}. \end{array}\right.$$

In our algorithm, we set the threshold parameter *t* following the above condition.

**Lemma** **2**([17]). *Given λ>0, $u\in \left(0,\frac{1}{{∥\Phi \Psi ∥}_{2}^{2}}\right)$, if $x^{*}$ is an optimal solution of $\left(P_{P_{a}}^{\lambda }\right)$, then $x^{*}$ is a fixed point of the threshold operator as (Equation 18).*

The above two lemmas will be used in the convergence proof of the algorithm. Next, we defined several auxiliary functions to prove convergence.
(28)$$\begin{array}{cc} R\left(x_{0},x_{s}\right):= & {∥\Phi \Psi x_{0}+\Phi x_{s}-y∥}_{2}^{2}\\ & +\lambda_{1}P_{a}\left(x_{0}\right)+\lambda_{2}P_{a}\left(x_{s}\right) \end{array}$$
(29)$$\begin{array}{cc} B_{x_{s}}\left(x_{0},v\right):= & u_{1}\left({∥\Phi \Psi x_{0}-\left(y-\Phi x_{s}\right)∥}_{2}^{2}\right.\\ \; & +\lambda_{1}P_{a}\left(x_{0}\right)+\lambda_{2}P_{a}\left(x_{s}\right)\\ \; & \left.-{∥\Phi \Psi x_{0}-\Phi \Psi v∥}_{2}^{2}\right)+{∥x_{0}-v∥}_{2}^{2} \end{array}$$
(30)$$\begin{array}{cc} C_{x_{0}}\left(x_{s},v\right):= & u_{2}\left({∥\Phi x_{s}-\left(y-\Phi \Psi x_{0}\right)∥}_{2}^{2}\right.\\ \; & +\lambda_{1}P_{a}\left(x_{0}\right)+\lambda_{2}P_{a}\left(x_{s}\right)\\ \; & \left.-{∥\Phi x_{s}-\Phi v∥}_{2}^{2}\right)+{∥x_{s}-v∥}_{2}^{2} \end{array}$$

**Theorem** **1.**
*Suppose that $x_{0}^{*}$ is a minimizer of of $B_{x_{s}}\left(x_{0},v\right)$ for any fixed $\lambda_{1}$, $\lambda_{2}$, $u_{1}$, $x_{s}$, and v. Then,*

(31)
$${\left(x_{0}^{*}\right)}_{i}=H_{\lambda_{1}u_{1},t}\left(z_{u_{1}}\left(v_{i}\right)\right).$$



**Proof.** Note that $B_{x_{s}}\left(x_{0},v\right)$ can be written as
$$\begin{array}{cc} B_{x_{s}}\left(x_{0},v\right)= & u_{1}\left({∥\Phi \Psi x_{0}-\left(y-\Phi x_{s}\right)∥}_{2}^{2}\right.\\ & +\lambda_{1}P_{a}\left(x_{0}\right)+\lambda_{2}P_{a}\left(x_{s}\right)\\ & \left.-{∥\Phi \Psi x_{0}-\Phi \Psi v∥}_{2}^{2}\right)+{∥x_{0}-v∥}_{2}^{2}\\ = & u_{1}{∥y-\Phi x_{s}∥}_{2}^{2}-2u_{1}\langle \Phi \Psi x_{0},y-\Phi x_{s}\rangle \\ & +\lambda_{1}u_{1}P_{a}\left(x_{0}\right)+\lambda_{2}u_{1}P_{a}\left(x_{s}\right)\\ & -u_{1}{∥\Phi \Psi v∥}_{2}^{2}+2u_{1}\langle \Phi \Psi x_{0},\Phi \Psi v\rangle \\ & +{∥x_{0}∥}_{2}^{2}+{∥v∥}_{2}^{2}-2\langle x_{0},v\rangle \\ = & {∥x_{0}∥}_{2}^{2}-2\langle x_{0},z_{u_{1}}\left(v\right)\rangle \\ & +\lambda_{1}u_{1}P_{a}\left(x_{0}\right)+\lambda_{2}u_{1}P_{a}\left(x_{s}\right)\\ & +{∥v∥}_{2}^{2}+u_{1}{∥y-\Phi x_{s}∥}_{2}^{2}-u_{1}{∥\Phi \Psi v∥}_{2}^{2}\\ = & {∥x_{0}-z_{u_{1}}\left(v\right)∥}_{2}^{2}+\lambda_{1}u_{1}P_{a}\left(x_{0}\right)\\ & +\lambda_{2}u_{1}P_{a}\left(x_{s}\right)+{∥v∥}_{2}^{2}+u_{1}{∥y-\Phi x_{s}∥}_{2}^{2}\\ & -u_{1}{∥\Phi \Psi v∥}_{2}^{2}-{∥z_{u_{1}}\left(v\right)∥}_{2}^{2}. \end{array}$$Because $\lambda_{1}$, $\lambda_{2}$, $u_{1}$, $x_{s}$, and *v* are fixed, using Lemma 1 gives us
$${\left(x_{0}^{*}\right)}_{i}=H_{\lambda_{1}u_{1},t}\left(z_{u_{1}}\left(v_{i}\right)\right).$$□

**Theorem** **2.**
*Suppose that $x_{s}^{*}$ is a minimizer of $C_{x_{0}}\left(x_{s},v\right)$ for any fixed $\lambda_{1}$, $\lambda_{2}$, $u_{2}$, $x_{0}$, and v. Then,*

(32)
$${\left(x_{s}^{*}\right)}_{i}=H_{\lambda_{2}u_{2},t}\left(z_{u_{2}}\left(v_{i}\right)\right).$$



**Proof.** It is similar to Theorem 1, so it was omitted. □

**Theorem** **3.**
*Let $u_{1}\in \left(0,\frac{1}{{∥\Phi \Psi ∥}_{2}^{2}}\right)$. Given that $\lambda_{1}$, $\lambda_{2}>0$ for any fixed $x_{s}$, suppose that $x_{0}^{*}$ is the global minimizer of $R\left(x_{0},x_{s}\right)$. Then, for any $x\in R^{N}$, we have*

(33)
$$B_{x_{s}}\left(x_{0}^{*},x_{0}^{*}\right)\leq B_{x_{s}}\left(x,x_{0}^{*}\right).$$



**Proof.** Because of the properties of the operator norm, for any $x\in R^{N}$, the following inequality holds:
$$∥\Phi \Psi x_{0}-\Phi \Psi x_{0}^{*}∥\leq {∥\Phi \Psi ∥}_{2}^{2}{∥x-x_{0}^{*}∥}_{2}^{2}\leq \frac{1}{u_{1}}{∥x-x_{0}^{*}∥}_{2}^{2}$$Hence, for any *x*,
$$\begin{array}{cc} B_{x_{s}}\left(x_{0}^{*},x_{0}^{*}\right)= & u_{1}R\left(x_{0}^{*},x_{s}\right)\leq u_{1}R\left(x,x_{s}\right)\\ = & u_{1}({∥\Phi \Psi x-\left(y-\Phi x_{s}\right)∥}_{2}^{2}\\ & \left.\;+\lambda_{1}P_{a}\left(x\right)+\lambda_{2}P_{a}\left(x_{s}\right)\right)\\ \leq & u_{1}({∥\Phi \Psi x-\left(y-\Phi x_{s}\right)∥}_{2}^{2}\\ & \left.+\lambda_{1}P_{a}\left(x\right)+\lambda_{2}P_{a}\left(x_{s}\right)\right)\\ & -u_{1}{∥\Phi \Psi x-\Phi \Psi x_{0}^{*}∥}_{2}^{2}\\ & +{∥x-x_{0}^{*}∥}_{2}^{2}\\ = & B_{x_{s}}\left(x,x_{0}^{*}\right) \end{array}$$□

**Theorem** **4.**
*Let $u_{2}\in \left(0,\frac{1}{{∥\Phi ∥}_{2}^{2}}\right)$. Given that $\lambda_{1}$, $\lambda_{2}>0$, for any fixed $x_{0}$, suppose that $x_{s}^{*}$ is the global minimizer of $R\left(x_{0},x_{s}\right)$. Then, for any $x\in R^{N}$, we have*

(34)
$$C_{x_{0}}\left(x_{s}^{*},x_{s}^{*}\right)\leq C_{x_{0}}\left(x,x_{s}^{*}\right).$$



**Proof.** It is similar to Theorem 3, so it was omitted. □

Now, we can derive conclusions about the convergence of the AD-IFPT algorithm.

**Theorem** **5.**
*Given $\lambda_{1}$, $\lambda_{2}>0$; $u_{1}\in \left(0,\frac{1}{{∥\Phi \Psi ∥}_{2}^{2}}\right)$, and $u_{2}\in \left(0,\frac{1}{{∥\Phi ∥}_{2}^{2}}\right)$, let sequence $\left\{x_{0}^{k}\right\}$, $\left\{x_{s}^{k}\right\}$ be generated by the form (Equation 24) and (Equation 25). Then, the following propositions hold:*

*(i) $\left\{R(x_{0}^{k},x_{s}^{k})\right\}$ is a monotonically decreasing sequence associated with k.*

*(ii) $\lim_{k\to \infty }{∥x_{0}^{k+1}-x_{0}^{k}∥}_{2}=0$ and $\lim_{k\to \infty }{∥x_{s}^{k+1}-x_{s}^{k}∥}_{2}=0$.*


**Proof.** (i) According to Theorem 1 and 3, we derived
$$\begin{array}{cc} & \;\;\;\;\;\min_{x}B_{x_{s}^{k}}\left(x,x_{0}^{k}\right)\\ & =B_{x_{s}^{k}}\left(H_{\lambda_{1}u_{1},t}\left(x_{0}^{k}+u_{1}{\left(\Phi \Psi \right)}^{T}\left(y-\Phi x_{s}^{k}-\Phi \Psi x_{0}^{k}\right)\right),x_{0}^{k}\right)\\ & =B_{x_{s}^{k}}\left(x_{0}^{k+1},x_{0}^{k}\right). \end{array}$$Similarly, we had
$$\begin{array}{c} \min_{x}C_{x_{0}^{k}}\left(x,x_{s}^{k}\right)=C_{x_{0}^{k}}\left(x_{s}^{k+1},x_{s}^{k}\right); \end{array}$$
hence,
$$\begin{array}{cc} R\left(x_{0}^{k+1}\right. & ,\left.x_{s}^{k+1}\right)\\ = & \frac{1}{u_{2}}\left(C_{x_{0}^{k+1}}\left(x_{s}^{k+1},x_{s}^{k}\right)-{∥x_{s}^{k+1}-x_{s}^{k}∥}_{2}^{2}\right)\\ & +{∥\Phi x_{s}^{k+1}-\Phi x_{s}^{k}∥}_{2}^{2}\\ \leq & \frac{1}{u_{2}}C_{x_{0}^{k+1}}\left(x_{s}^{k},x_{s}^{k}\right)+{∥\Phi x_{s}^{k+1}-\Phi x_{s}^{k}∥}_{2}^{2}\\ & -\frac{1}{u_{2}}{∥x_{s}^{k+1}-x_{s}^{k}∥}_{2}^{2}\\ \leq & \frac{1}{u_{2}}C_{x_{0}^{k+1}}\left(x_{s}^{k},x_{s}^{k}\right)\\ = & R\left(x_{0}^{k+1},x_{s}^{k}\right)\\ = & \frac{1}{u_{1}}\left(B_{x_{s}^{k}}\left(x_{0}^{k+1},x_{0}^{k}\right)-{∥x_{0}^{k+1}-x_{0}^{k}∥}_{2}^{2}\right)\\ & +{∥\Phi \Psi x_{0}^{k+1}-\Phi \Psi x_{0}^{k}∥}_{2}^{2}\\ \leq & \frac{1}{u_{1}}B_{x_{s}^{k}}\left(x_{0}^{k},x_{0}^{k}\right)-{∥\Phi \Psi x_{0}^{k+1}-\Phi \Psi x_{0}^{k}∥}_{2}^{2}\\ & -\frac{1}{u_{1}}{∥x_{0}^{k+1}-x_{0}^{k}∥}_{2}^{2}\\ \leq & \frac{1}{u_{1}}B_{x_{s}^{k}}\left(x_{0}^{k},x_{0}^{k}\right)\\ = & R\left(x_{0}^{k},x_{s}^{k}\right), \end{array}$$
which shows that $\left\{R(x_{0}^{k},x_{s}^{k})\right\}$ is a monotonically decreasing sequence associated with *k*.(ii) Using the properties of the operator norm and Theorem 3, we had
$$\begin{array}{cc} & u_{1}\left(R\left(x_{0}^{k},x_{s}^{k}\right)-R\left(x_{0}^{k+1},x_{s}^{k+1}\right)\right)\\ = & B_{x_{s}^{k}}\left(x_{0}^{k},x_{s}^{k}\right)-u_{1}R\left(x_{0}^{k+1},x_{s}^{k+1}\right)\\ \geq & B_{x_{s}^{k}}\left(x_{0}^{k+1},x_{0}^{k}\right)-u_{1}R\left(x_{0}^{k+1},x_{s}^{k+1}\right)\\ = & {∥x_{0}^{k+1}-x_{0}^{k}∥}_{2}^{2}-u_{1}{∥\Phi \Psi x_{0}^{k+1}-\Phi \Psi x_{0}^{k}∥}_{2}^{2}\\ & +u_{1}\left({∥\Phi \Psi x_{0}^{k+1}-\left(y-\Phi x_{s}^{k}\right)∥}_{2}^{2}+\lambda_{2}P_{a}\left(x_{s}^{k}\right)\right.\\ & -∥{\Phi \Psi x_{0}^{k+1}-\left(y-\Phi x_{s}^{k}\right)∥}_{2}^{2}-\lambda_{2}P_{a}\left(x_{s}^{k+1}\right))\\ \geq & {∥x_{0}^{k+1}-x_{0}^{k}∥}_{2}^{2}-u_{1}{∥\Phi \Psi x_{0}^{k+1}-\Phi \Psi x_{0}^{k}∥}_{2}^{2}\\ \geq & {∥x_{0}^{k+1}-x_{0}^{k}∥}_{2}^{2}-u_{1}{∥\Phi \Psi ∥}_{2}^{2}{∥x_{0}^{k+1}-x_{0}^{k}∥}_{2}^{2}\\ = & \left(1-u_{1}{∥\Phi \Psi ∥}_{2}^{2}\right){∥x_{0}^{k+1}-x_{0}^{k}∥}_{2}^{2} \end{array}$$According to Theorem 4, and using a similar method, we derived
$$\begin{array}{cc} & u_{2}\left(R\left(x_{0}^{k},x_{s}^{k}\right)-R\left(x_{0}^{k+1},x_{s}^{k+1}\right)\right)\\ \geq & \left(1-u_{2}{∥\Phi ∥}_{2}^{2}\right){∥x_{s}^{k+1}-x_{s}^{k}∥}_{2}^{2}. \end{array}$$Hence, we discovered that
$$\begin{array}{cc} \sum_{k=0}^{N}{∥x_{0}^{k+1}-x_{0}^{k}∥}_{2}^{2} & \leq \frac{u_{1}\left(R\left(x_{0}^{0},x_{s}^{0}\right)-R\left(x_{0}^{N},x_{s}^{N}\right)\right)}{1-u_{1}{∥\Phi \Psi ∥}_{2}^{2}}\\ & \leq \frac{u_{1}R\left(x_{0}^{0},x_{s}^{0}\right)}{1-u_{1}{∥\Phi \Psi ∥}_{2}^{2}}\\ & =\frac{u_{1}{∥y∥}_{2}^{2}}{1-u_{1}{∥\Phi \Psi ∥}_{2}^{2}} \end{array}$$
and
$$\begin{array}{cc} \sum_{k=0}^{N}{∥x_{s}^{k+1}-x_{s}^{k}∥}_{2}^{2} & \leq \frac{u_{2}\left(R\left(x_{0}^{0},x_{s}^{0}\right)-R\left(x_{0}^{N},x_{s}^{N}\right)\right)}{1-u_{2}{∥\Phi ∥}_{2}^{2}}\\ & \leq \frac{u_{2}R\left(x_{0}^{0},x_{s}^{0}\right)}{1-u_{2}{∥\Phi ∥}_{2}^{2}}\\ & =\frac{u_{2}{∥y∥}_{2}^{2}}{1-u_{2}{∥\Phi ∥}_{2}^{2}} \end{array}$$Notice that to the right of the above two inequalities are constant values. Let N→+∞; it followed immediately that these two series were convergent, and this gave us
$$\lim_{k\to \infty }{∥x_{0}^{k+1}-x_{0}^{k}∥}_{2}=0$$
and
$$\lim_{k\to \infty }{∥x_{s}^{k+1}-x_{s}^{k}∥}_{2}=0$$□

## 4. Performance Evaluation and Experimental Results

In this part, we first introduce an energy consumption model of the WSN, which evaluated power consumption by the number of instructions executed on the hardware. Then, we evaluated the energy efficiency performance of the proposed non-convex fraction function compressed sensing model. Furthermore, we compared the energy consumption of the IFPT and AD-IFPT with other traditional methods on the WSN during abnormal events’ monitoring.

### 4.1. Energy Consumption Estimation Model of the WSN

In our analysis, we used the energy consumption model of Mica2 [32] to describe all the energy consumption of our models. Mica2 is equipped with an Atmel Atmega 128L processor and a Chipcon CC1000 radio. We took into consideration the energy consumption caused by the hardware instruction execution times of different methods under a fixed hardware setup and transmission distance. Assuming that the distance between each node was 100 m, the hardware parameters of the sensor energy consumption are shown in Table 1, as given by [32]. The energy consumption at the node was expressed as follows:(35)$$E=E_{REC}+E_{SMP}+E_{BG}+E_{COM}+E_{TSM},$$
where $E_{REC}$ is the energy consumption of receiving signal; $E_{SMP}$ is the energy consumption of signal sampling; $E_{BG}$ is the energy consumption of background power consumption during hardware operation; $E_{BG}$ is the background energy consumption of the hardware: $E_{TSM}$ is the energy consumption of a transmitting signal; $E_{COM}$ is the energy of computation in the hardware, which depends on the kind of processing and operating carried out for the signal on the nodes.

In this paper, two working modes of compressed sensing were analyzed: analog compressed sensing and compressed data gathering in a multi-hop WSN.

#### 4.1.1. Energy Consumption of Analog Compressed Sensing

In this case, we assumed that the length of the signal was *N* under Nyquist sampling. When using compressed sensing, we sampled the original signal *M* times with M≪N. For processor instruction, there were *M* times of inner product computing of *N*-dimensional vectors under the analog compressed sensing mode we introduced in Section 2. We set the random sequence to satisfy the Bernoulli distribution, which takes the value of 0 or 1 with a 0.5 probability, respectively. Therefore, the energy consumption calculation formula of this case is given by
(36)$$\begin{array}{cc} E_{\left(trad\right)}= & E_{REC}+E_{SMP}+E_{BG}+E_{COM}+E_{TSM}\\ = & \left(NE_{rx}\right)+\left(NP_{acq}D_{op}\right)+\left(2NP_{bg}D_{op}\right)\\ & +\left(N\varepsilon_{mrd}+N\varepsilon_{mwr}\right)+\left(NE_{tx}\right), \end{array}$$
(37)$$\begin{array}{cc} E_{\left(CS\right)}= & E_{REC}+E_{SMP}+E_{BG}+E_{COM}+E_{TSM}\\ = & \left(NE_{rx}\right)+\left(NP_{acq}D_{op}\right)\\ & +M\left[\left(M+N+MN\right)P_{bg}D_{op}\right]\\ & +\left(N\varepsilon_{mrd}+MN\varepsilon_{add}+M\varepsilon_{mwr}\right)\\ & +\left(ME_{tx}\right), \end{array}$$
where $E_{\left(trad\right)}$ represents the energy consumption of the WSN in traditional mode and $E_{\left(CS\right)}$ represents its energy consumption in compressed sensing mode.

#### 4.1.2. Energy Consumption of CDG in Multi-Hop WSN

In this case, we assumed that there were *N* nodes in a signal transmission chain of a multi-hop WSN. In the traditional mode, each node generates a reading from its sampling, receives the readings of all upstream nodes, and transmits them to the next node. Consequently, there were N×(N+1)/2 times of data transmission in the overall process. In the compressed sensing mode, each node receives a binary number (0 or 1), which determines whether or not it was sampled. If a node received a 1, it generated a reading in the sampling, received a number from the nearest upstream node, and then, transmitted their sum to the next node. If a node received a 0, it only had to transmit a number from the nearest upstream node to the next one without sampling. This step was repeated *M* times. Therefore, the energy consumption calculation formula of this case is given by:(38)$$\begin{array}{cc} E_{\left(trad\right)}=\; & E_{REC}+E_{SMP}+E_{BG}+E_{COM}+E_{TSM}\\ =\; & \left(\frac{1}{2}N\left(N-1\right)E_{rx}\right)+\left(NP_{acq}D_{op}\right)\\ & +\left(N(N-1)+2N\right)P_{bg}D_{op}\\ & +\left(\frac{1}{2}N\left(N+1\right)\left(\varepsilon_{mrd}+\varepsilon_{mwr}\right)\right)\\ & +\left(\frac{1}{2}N\left(N+1\right)E_{tx}\right), \end{array}$$
(39)$$\begin{array}{cc} E_{\left(CS\right)}=\; & E_{REC}+E_{SMP}+E_{BG}+E_{COM}+E_{TSM}\\ =\; & \left(MNE_{rx}\right)+\left(NP_{acq}D_{op}\right)\\ & +\left(4MN+M(N-1)\right)P_{bg}D_{op}\\ & +\left(2MN\varepsilon_{mrd}+M\left(N-1\right)\varepsilon_{add}+2MN\varepsilon_{mwr}\right)\\ & +\left(MNE_{tx}\right), \end{array}$$
where $E_{\left(trad\right)}$ represents the energy consumption of the WSN in traditional mode and $E_{\left(CS\right)}$ represents it in compressed sensing mode.

### 4.2. Performance of the Proposed Model and Energy Consumption Comparison

In this part, we first evaluated the availability of the proposed AD-IFPT algorithm in abnormal events’ monitoring. We randomly generated signals with a length of 512, sparsity level $K_{1}$ = 64, abnormal events $K_{2}$ = 16, and −30 dB of Gaussian noise, then reconstructed the signals using 250 measurements by the AD-IFPT (see Figure 5). The result showed that the AD-IFT algorithm could reconstruct the original signal with noise and abnormal events within a 0.1% error range. Therefore, it is advisable to use the fraction penalty function model and the AD-IFPT algorithm to reconstruct the signal.

Next, we analyzed the energy consumption of the AD-IFPT, IFPT, and other traditional compressed sensing models. Before starting the analysis, it was necessary to define the criteria for the successful reconstruction of the signal. We used the normalize mean-squared error (NMSE) to evaluate the reconstruction error of signal *d* as
(40)$$\mathrm{NMSE}\left(d\right)=\frac{{∥d-d^{*}∥}_{2}^{2}}{{∥d^{*}∥}_{2}^{2}},$$
where $d^{*}$ represents the original signal and *d* represents the recovery signal. Let us assume that *d* is called a successful reconstructed signal when $\mathrm{NMSE}\left(d\right)\leq 10^{-3}$.

The following is the comparison of the AD-IFPT, IFPT, and two traditional compressed sensing models in the experiment. The last three algorithms adopted the matrix splicing method in [6] when reconstructing abnormal event monitoring signals. The primitive dual-interior point method based on $\ell_{1}$-norm relaxation (the built-in compressed sensing algorithm in the L1-MAGIC package in MATLAB) and the greedy algorithm were selected as the contrast.

The result showed that the IFPT and AD-IFPT had better performance compared to the traditional compressed sensing methods; more specifically, the number of measurements of the proposed models was significantly reduced. The AD-IFPT had the best performance: it stably reconstructed a 512-dimensional signal in 170 measurements without abnormal events and in 225 measurements with abnormal events (see Figure 6).

In the simulation experiment, we determined that, when each algorithm reconstruction rate reached 100% stability, the number of its measurements could be accepted. Moreover, our experiment was carried out at different sparsity signal levels (0.05 to 0.275) and considered two scenarios: WSN monitoring with and without abnormal events. In addition, we calculated energy consumption in two cases: analog compressed sensing and compressed data gathering in the multi-hop WSN (see Figure 7). The results showed that, compared with transmitting the signal directly, the energy consumption of the compressed sensing method was significantly reduced when the signal sparsity decreased, and the energy savings reached more than 50% of the direct transmission when K/N was below 0.05. In the comparison with the compressed sensing methods, the AD-IFPT reduced the energy consumption compared with other traditional compressed sensing models at the same sparsity level.

**Remark** **1.**
*It is worth noting that, in each iteration, the time complexity of the IFPT algorithm was O(MN), mainly caused by the multiplication of the matrix and vector. Assuming that it takes T iterations to finish the algorithm, then the time complexity of our algorithm is O(TMN), where T is related to the parameters. The complexity of the algorithms compared in this paper, i.e., convex optimization and the greedy algorithm, was $O(N^{3})$ and O(KMN), respectively [33].*


## 5. Conclusions

In this paper, we mainly discussed the energy consumption of the sensor nodes in a WSN by using a non-convex compressed sensing model, which is different from the current popular models where the $\ell_{1}$-norm is used as the surrogate function of the $\ell_{0}$-norm. However, we adopted a specific non-convex surrogate function (the fraction function) as the surrogate function of the $\ell_{0}$-norm. For abnormal events’ monitoring by the WSN, considering the sparse characteristics of abnormal events, we proposed an optimization model and designed an AD-IFPT algorithm to solve this optimization model, which avoided the extra measurements of previous algorithms. Finally, we used a WSN energy consumption model for the sensor nodes in the WSN under different methods. The results showed that our model and algorithm reduced the amount of data acquisition and transmission such that the energy consumption of the sensor nodes was effectively reduced.

## Figures and Tables

**Figure 1 sensors-22-08378-f001:**
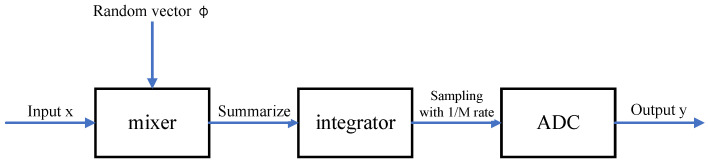
A classical random demodulator (RD) architecture in an analog compressed sensing application.

**Figure 2 sensors-22-08378-f002:**
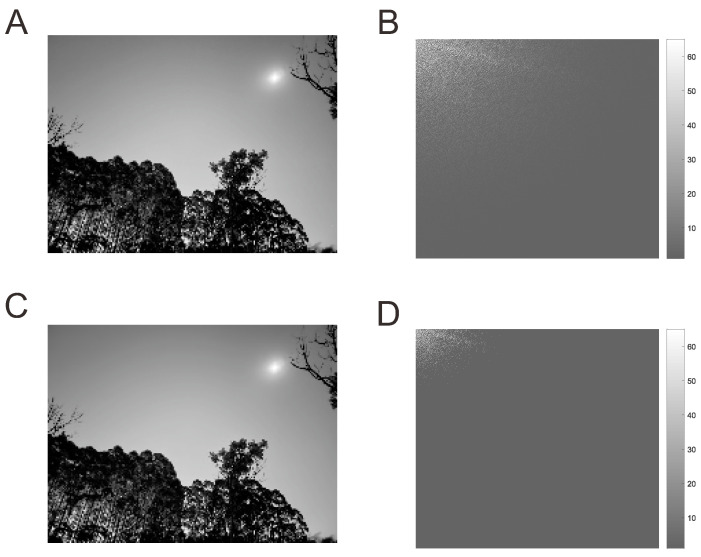
(**A**) Original image; (**B**) coefficient matrix under discrete cosine transform (DCT); (**C**) compressed image using 5% largest coefficient; (**D**) the 5% largest coefficient matrix.

**Figure 3 sensors-22-08378-f003:**
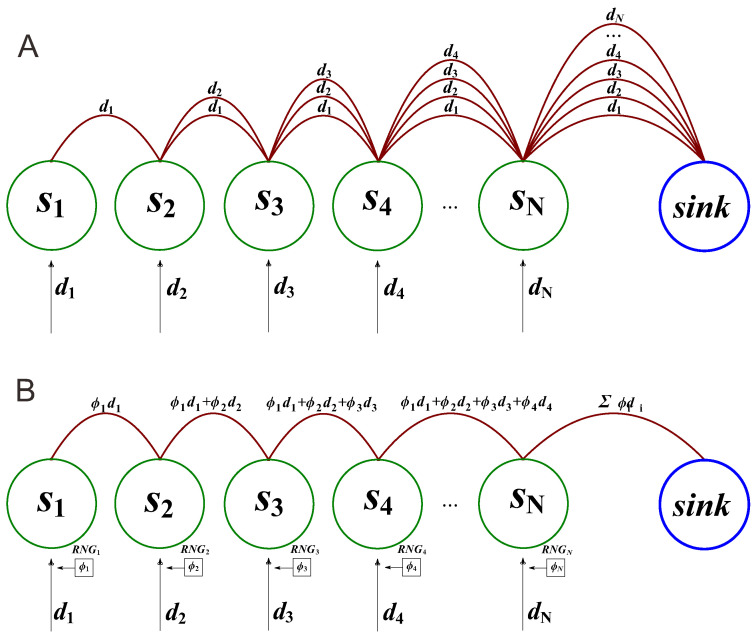
(**A**) Traditional transmitting method of a multi-hop WSN; (**B**) CS method of a multi-hop WSN.

**Figure 4 sensors-22-08378-f004:**
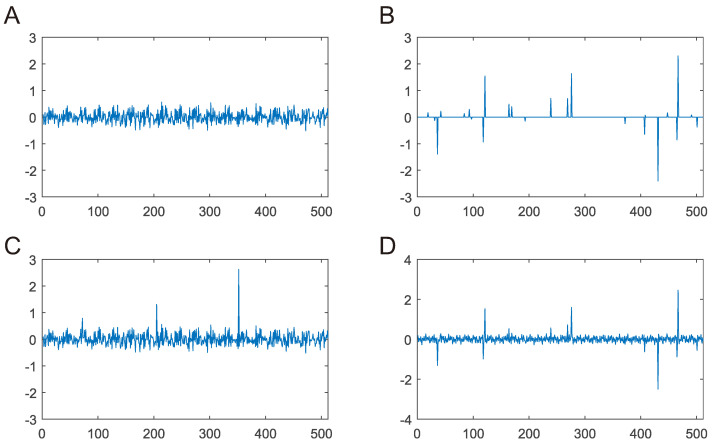
(**A**) Original signal (*N* = 512); (**B**) signal in sparse domain (*K* = 25); (**C**) signal with abnormal events; (**D**) signal with abnormal events in sparse domain.

**Figure 5 sensors-22-08378-f005:**
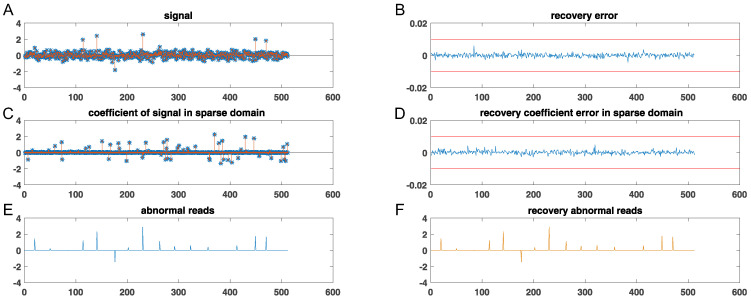
(**A**) Original signal and recovery signal; (**B**) recovery error of reconstruction; (**C**) reconstruction sparse domain coefficient; (**D**) recovery error of sparse domain coefficient; (**E**) abnormal events of a wireless sensor network; (**F**) the recovery result of abnormal events.

**Figure 6 sensors-22-08378-f006:**
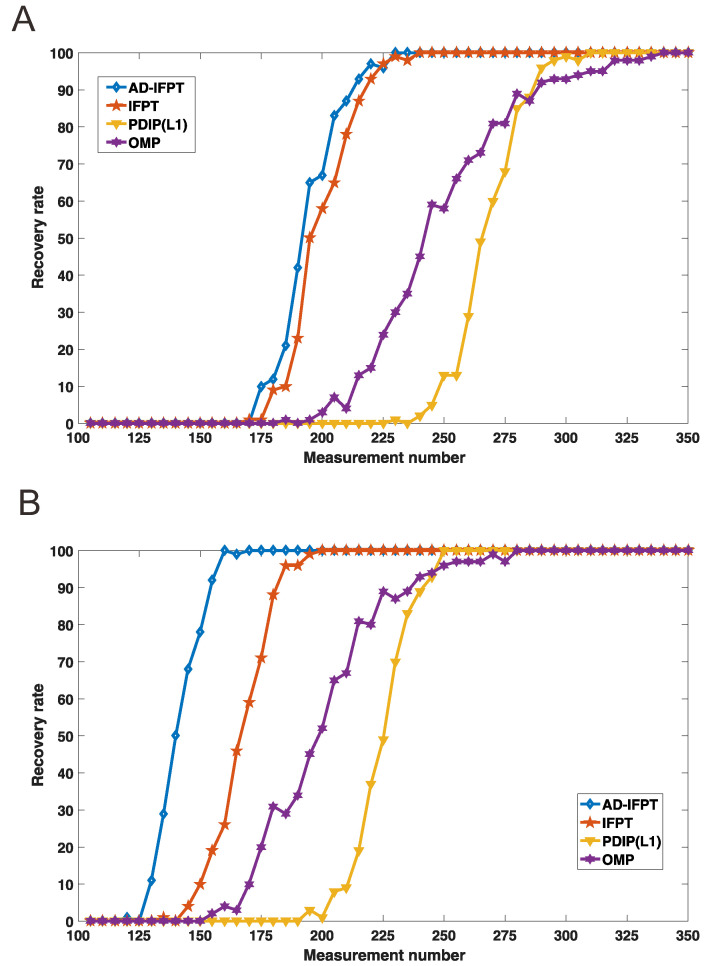
(**A**) Comparison of the recovery rate of 4 algorithms with abnormal events; (**B**) comparison of the recovery rate of 4 algorithms without abnormal events.

**Figure 7 sensors-22-08378-f007:**
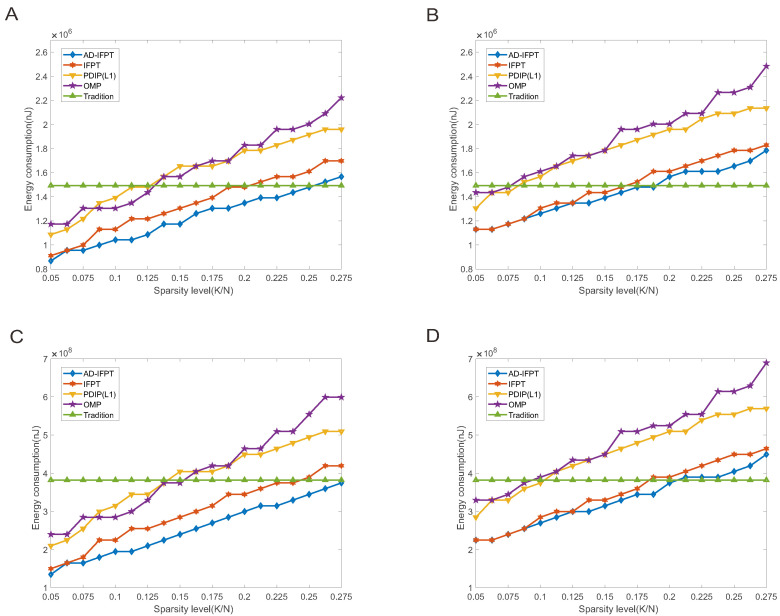
Energy consumption in four cases. (**A**) Analog compressed sensing processing; (**B**) analog compressed sensing with abnormal events; (**C**) compressed data gathering in a multi-hop WSN; (**D**) compressed data gathering in a multi-hop WSN with abnormal events.

**Table 1 sensors-22-08378-t001:** Hardware parameters.

Parameter	Value	Description
$P_{acq}$	15.01 mW	Signal sampling power consumption
$E_{rx}$	0.923 μJ	Energy consumption for receiving signals
$E_{tx}$	1.984 μJ	Energy consumption for transmitting signals
$D_{op}$	0.14 μs	Duration of a CPU operation
$P_{bg}$	9.6 mW	Hardware background power consumption
$\varepsilon_{add}$	3.30 nJ	Energy consumption for addition operation
$\varepsilon_{mul}$	9.90 nJ	Energy consumption for multiplication operation
$\varepsilon_{mrd}$	0.26 nJ	Energy consumption for memory reading
$\varepsilon_{mwr}$	4.30 nJ	Energy consumption for memory writing

## Data Availability

Not applicable.

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
