# Peer review of "A Non-Convex Compressed Sensing Model Improving the Energy Efficiency of WSNs for Abnormal Events’ Monitoring"

_sensors, 2022, doi:10.3390/s22218378_

Round 1

Reviewer 1 Report

The paper mainly presents two key contributions:

                  1.            The proposition of an optimization compressed - sensing model in the wireless sensor network (WSN) based on the non-convex fraction function and an alternate direction iterative fraction penalty thresholding AD-IFPT algorithm to solve it.

                  2.            The comparison of the WSN energy consumption in different cases between the AD-IFPT, IFPT and two traditional compressed sensing models.

This paper focuses on a compressed sensing-based WSN reconstruction model applying non-convex approach. The non-convex approach replaces â„“1-norm by â„“p-norm, where, 0 < p < 1. This approach is able to recover the sparse solution from much fewer measurements compared to the convex approach. Another advantage of non-convex approach is that a weaker version of restricted isometry property (RIP) condition is sufficient for perfect reconstruction. Moreover, the number of measurements and error decreases with p.

However, the disadvantages of this reconstruction model are slower and complex algorithm and difficulties in implementation for problems of large size. The authors conducted experiments with 170, 225 and 250 measurements. Have the authors considered this problem and compared the proposed algorithm with others in terms of computational burden? What about the time of reconstruction algorithm execution. Please clarify this.

In presented paper, the authors discussed effectiveness of the energy consumption of the sensor nodes in WSN using proposed reconstruction algorithm. The results are promising and energy consumption during abnormal events are reduced.

The literature cited is relevant to the study but requires supplementing with items that have appeared in recent years and relate to this topic.

In general, I think that the topic addressed in this study should be of interest to many of the wide readership of the journal Sensors. The findings are interesting, and the research design and methodology seem proper to me.

The paper in its present form, with improvements suggested in the previous section, makes an acceptable case for publication.

Author Response

Thank you for your careful analysis and suggestions on our work. Your opinions are right. We test the time consuming of the algorithm in experiment, and the results show that proposed model, convex relaxation model and greedy model take 0.21 seconds, 0.13 seconds and 0.07 seconds respectively (repeated 50 times and used average values) under the parameters M=300, K=80 and N=512. The reason for the high time consumption in our experiment may be that the threshold algorithm of non-convex function requires much more steps to converge. Moreover, we add the runtime burden of the algorithm and corresponding references as a remark at the page 18, line 264.

Remark 1. It is worth noting that, in each iteration, the time complexity of IFPT algorithm is O(MN), mainly caused by the multiplication of matrix and vector. Assuming that it takes T iterations to finish the algorithm, then the time complexity of our algorithm is O(TMN), where T is related to the parameters. The complexity of the algorithms compared in this paper, i.e. convex optimization and greedy algorithm, is O(N^3) and O(KMN) respectively [33]. ”

Because our paper considers that the reconstruction algorithm runs at the sink in WSN and does not strongly require real-time performance, the time of reconstruction algorithm execution is acceptable.

Reviewer 2 Report

This paper proposed a new sparsity promoted surrogate function for compressive sensing. The manuscript is well-organized and technically sounds. My minor suggestion is that some other smoothed surrogate functions should be compared in both noiseless and noisy situation. 

Author Response

Thank you for your careful analysis and suggestions on our work. We add the relevant explanation at the page 7, line 157.

“The comparison of fraction penalty function with another non-convex function â„“-0.5 norm have been explained in [17] in detail, which shows that the fraction penalty function has better sparse approximation and lower number of measurements than â„“-0.5 norm. Better properties of fraction penalty function are helpful to achieve better energy efficiency in WSN.”

It should be noted that we do not compare the fractional function with other non-convex and non-smooth surrogate functions such as â„“-p norm (p not equal 0.5) and SCAD etc., which mainly because that, in our paper, we focus on the performance between the non-convex model and convex model in WSN and abnormal events monitoring.